

# Earthquake ruptures and topography controlled by plate interface deformation

Nadaya Cubas[1], Philippe Agard[1], and Roxane Tissandier[2]

[1]Sorbonne Université, CNRS-INSU, Institut des Sciences de la Terre Paris, ISTeP UMR 7193, F-75005 Paris, France
[2]Institut de Physique du Globe de Paris, Université de Paris, CNRS, 75238 Paris, France

**Correspondence:** Nadaya Cubas (nadaya.cubas@sorbonne-universite.fr)

**Abstract.** What controls the location and segmentation of mega-earthquakes in subduction zones is a long-standing problem in earth sciences. Prediction of earthquake ruptures mostly relies on interplate coupling models based on Global Navigation Satellite Systems providing patterns of slip deficit between tectonic plates. We here investigate if and how the seismic and aseismic patches revealed by these models relate to the distribution of deformation along the plate interface, i.e. basal erosion

and/or underplating. From a mechanical analysis of the topography applied along the Chilean subduction zone, we show that extensive plate interface deformation takes place along most of the margin. We show that basal erosion occurs preferentially at 15 km depth while underplating does at $35 \pm 10$ and $60 \pm 5$ km depth, in agreement with P-T conditions of recovered underplated material, expected pore pressures, and spatial distribution of marine terraces and uplift rates. Along southern Chile, large sediment input favors shallow accretion and underplating of subducted sediments, while along northern Chile,

extensive basal erosion provides material for the underplating. We then show that all major earthquakes of southern Chile are limited along their down-dip end by underplating while, along northern Chile, they are surrounded by both basal erosion and underplating. Segments with heterogeneously distributed deformation largely coincide with lateral earthquake terminations. We therefore propose that long-lived plate interface deformation promotes stress build-up and leads to earthquake nucleation. Earthquakes then propagate along fault planes shielded from this long-lived permanent deformation, and are finally stopped

by segments of heterogeneously distributed deformation. Slip deficit patterns and earthquake segmentation therefore reflect the along-dip and along-strike distribution of the plate interface deformation. Topography acts as a mirror of distributed plate interface deformation and should be studied systematically to improve the prediction of earthquake ruptures.

## 1    Introduction

Predicting the extent of subduction zone seismic ruptures mostly relies on geodetic observations of interseismic coupling (e.g.,

Moreno et al., 2010), notwithstanding spatial resolution issues (Loveless and Meade, 2011). Some earthquakes have however ruptured low coupled zones (Noda and Lapusta, 2013) or only part of highly locked patches (e.g., Konca et al., 2008). This was the case for the 2014 Mw 8.4 Iquique earthquake, which only ruptured the northern end of the North Chile gap (last broken in 1877, Ruiz et al., 2014a), or the Tocopilla Mw 7.8 earthquake, which struck the down-dip and less coupled part of the same



gap seven years before (Delouis et al., 1998; Metois et al., 2016) (Fig. 1). Predicting the location and extent of earthquakes
clearly requires a better understanding of earthquake mechanics.

Subduction earthquake propagation has so far been either related to megathrust frictional properties (Perfettini et al., 2010;
Kaneko et al., 2010), inherited stress states (Konca et al., 2008; Kaneko et al., 2010) or seafloor roughness (Kodaira et al.,
2000; Wang and Bilek, 2014). Megathrusts, commonly regarded as interfingered seismic and aseismic patches with contrasting
frictional properties, are conventionally sub-divided into four domains with depth (A-D, Lay et al., 2012). The highly coupled
domain-B hosts mega-earthquakes, while the partly coupled domain-C is characterized by moderate slip earthquakes generating
significant high frequency seismic radiation (Lay et al., 2012).

The distribution of these patches might persist over time scales up to the million years and control forearc morphology
(Song and Simons, 2003). A spatial correlation between the coastline and the down-dip limit of the strongly locked zone has
indeed been evidenced along the northern chilean margin (Béjar-Pizarro et al., 2013; Metois et al., 2016). Coastal uplift rates
inferred from marine terraces, between 0.1 and 0.3 mm/a over the last 400 ka (Saillard et al., 2017), have been linked to the
change of mechanical coupling between domains B and C (Saillard et al., 2017) or to underplating (Delouis et al., 1998; Adam
and Reuther, 2000; Clift and Hartley, 2007). Numerical investigations recently suggested that underplating, through transient
stripping of the slab-top near the base of the forearc crust, may generate periodic Myr-long uplift sequences and a trackable
100 m-high topographic signal (Menant et al., 2020).

Bathymetric features are considered as potential seismic barriers, based on the distribution of incoming plate roughness
(Wang and Bilek, 2014; Lallemand et al., 2018) or gravimetric anomalies (Bassett and Watts, 2015). Subduction of bathymetric
features induces large fracture networks, promoting removal of material from the bottom of the overriding plate (basal erosion,
Ranero and von Huene, 2000) and potentially characteristic topographic features (Dominguez et al., 1998; Collot et al., 2008;
Ruh et al., 2016). Hetereogeneities in the structure of the damaged plate interface and stress field, as a result, would favor
aseismic slip and impede the propagation of large ruptures (Wang and Bilek, 2014). Very few seismic surveys, however, have
imaged seamounts surrounding a large rupture (Kodaira et al., 2000; Geersen et al., 2015).

We herein explore an alternative explanation: rather than variations of frictional properties or of seafloor roughness, earth-
quake segmentation might relate to the distribution of deformation along the plate interface (Wang and Bilek, 2014), as ob-
served along strike-slip faults (Wesnousky, 2006). Both underplating and basal erosion require the redistribution of deformation
ultimately leading to the migration of the plate boundary (Vannucchi et al., 2012; Agard et al., 2018). We here show that the
location of such distributed deformation along the plate interface can be captured from a simple mechanical analysis of the
topography, since underplating and basal erosion both impact forearc morphology. This method is applied over a 2000km long
strech along the Chilean margin, known to transition from erosive to accretionary from north to south, as incoming sediment
thickness increases (Clift and Vannucchi, 2004; Clift and Hartley, 2007) (Fig. 1a). This margin is one of the best instrumented
subduction zones, along which five mega-earthquakes, including a domain-C event, have occurred during the last 25 years (Fig.
1c). We first map out the location of distributed deformation zones associated with basal erosion and underplating and compare
them with the megathrust segmentation proposed so far. We then discuss how distributed deformation along the plate interface
impacts and therefore can be used to constrain earthquake extent and mechanics.





## 2 Methodology

To do so, we applied the critical taper theory (CTT, Davis et al., 1983, Fig. 2a). The theory predicts the topographic slope of a wedge on the verge of failure as well as dips of its internal faults, from the frictional and pore fluid pressure properties of the wedge and megathrust (Dahlen, 1984). We used the solution of Dahlen (1984) for a non cohesive wedge, describing the critical taper as a function of the angle $\Psi_B$ formed by the maximum principal stress $\sigma_1$ and the base of the wedge and the angle $\Psi_0$ formed by $\sigma_1$ and the top of the wedge. The solution for the lower compressional branch is:

$$(\alpha + \beta)_c = \Psi_B - \Psi_0 \tag{1}$$

with

$$\Psi_B = \frac{1}{2} arcsin(\frac{sin\phi_b'}{sin\phi_b}) - \frac{1}{2}\phi_b' \, , \tag{2}$$

$$\Psi_0 = \frac{1}{2} arcsin(\frac{sin\alpha'}{sin\phi_{int}}) - \frac{1}{2}\alpha'. \tag{3}$$

The angles $\phi_{int}$ and $\phi_b$ are the internal and basal coefficients of friction defined as $\mu_{int} = tan\phi_{int}$ and $\mu_b = tan\phi_b$, and

$$\phi_b' = arctan\left[\left(\frac{1-\lambda_b}{1-\lambda}\right)tan\phi_b\right] \, , \tag{4}$$

$$\alpha' = arctan\left[\left(\frac{1-\rho_w/\rho}{1-\lambda}\right)tan\alpha\right] . \tag{5}$$

The internal and basal Hubbert-Rubbey fluid pressure ratios $\lambda$ and $\lambda_b$ are defined in (Davis et al., 1983) as:

$$\lambda = \frac{P - \rho_w gD}{|\sigma_z| - \rho_w gD} \, , \tag{6}$$

$$\lambda_b = \frac{P_b - \rho_w gD}{|\sigma_z| - \rho_w gD} \, , \tag{7}$$

where $\rho$ and $\rho_w$ are the wedge material and water densities and $D$ is the water depth. The solution is exact if $\lambda = \lambda_b$ and the approximation is valid for small tapers as used in this study (Wang et al., 2006).

The relationship between the topographic slope $\alpha$ and the slab dip $\beta$ forms an envelope separating different mechanical states (Fig. 2a). The wedge is at critical state if the taper formed by $\alpha$ and $\beta$ follows a critical envelope. In that case, activation of the megathrust requires internal faulting. In contrast, if the taper exceeds this critical limit, the wedge enters in a stable domain (in the sense of the CTT), where the only possible active fault is the megathrust.

Dynamic effective frictions of seismogenic zones have been shown to reach extremely low values ($\mu \sim 0.01 - 0.03$, Fulton

et al., 2013; Gao and Wang, 2014). As a consequence, a forearc above a seismogenic megathrust, with standard $\alpha$ and $\beta$, will



systematically fall in the CTT stable domain (Fig. 2a). In contrast, aseismic megathrusts are characterized by larger effective friction ($\mu \sim 0.1 - 0.15$, Cubas et al., 2013; Gao and Wang, 2014), allowing the wedge to reach critical conditions (Cubas et al., 2013) (Fig. 2a). Furthermore, at critical state, if the basal effective friction of the wedge approaches the internal effective friction, the dip of internal faults within the wedge will decrease and become parallel to the plate interface. A small difference

of friction will thus lead to deformation either by basal accretion (in the lower plate) or by basal erosion (in the upper plate), instead of standard accretion characterized by thrust faults reaching the surface (Dahlen, 1984) (Fig. 2a). If the basal effective friction reaches the internal one, a highly fractured forearc is even expected.

Following the method of (Cubas et al., 2013), we used ETOPO 1 for the bathymetric and topographic slope, and slab 2.0 for the slab dip (Hayes et al., 2018). We built swath profiles perpendicular to the trench every 0.1 longitudinal degree (Suppl.

Mat. Fig. 1), smoothed by a rectangular window, to get average topographic slopes and slab dips with their standard deviations necessary to the inversion procedure (Fig. 2, Suppl. Mat. Fig. 2, 5). Along these profiles, we selected segments parallel to critical envelops. For each of these segments, we retrieved, by inversion, probability density functions for the friction of the wedge, the pore pressure ratio of the wedge and the effective friction of the megathrust (Suppl. Mat. Fig. 3). The CTT inversion only provides the internal pore pressure, which can be considered as a lower bound for the megathrust pore pressure. Moreover,

close effective internal and basal frictions implies close internal and megathrust pore pressure. The inversion procedure is described in (Cubas et al., 2013). We only kept segments with extremely low misfits, and with values consistent with standard frictions (from 25 to 43$^o$ for $\phi_{int}$, Byerlee (1978), from 1 to 42.9$^o$ for $\phi_b^{eff}$, and from 0.35 to 0. 95 for $\lambda$). Sensitivity tests were run by Cubas et al. (2013), showing that an error of $\pm 5^o$ on $\beta$ implies a horizontal translation of the taper and a 3$^o$ variation for the effective basal friction, without affecting the critical state of the forearc. Best misfits for each parameter are provided in

Supplementary Material (Table 1 & Fig. 6).

## 3  Results

### 3.1  Locating plate interface deformation

Since we seek to capture distribution of deformation along the plate interface, we compared the internal and megathrust effective frictions and mapped the difference (Fig. 1b). A one degree difference implies a dip difference $< 10^o$ between the

megathrust and the forward-verging thrusts. This representation allows inferring areas prone to either standard accretion, or to basal erosion/accretion, the latter being herein defined as distributed deformation along the plate interface.

The transition from erosion to accretion is particularly well captured by the method despite its relative simplicity (Fig. 1a-b). North of 25$^o$S (i.e., at the transition between erosive and accretionary subduction), the difference of effective frictions is systematically lower than 2$^o$, and mostly lower than 1$^o$, which means that deformation is essentially located along the

plate interface. The difference increases notably south of the Juan Fernandez ridge, and progresses inland towards the south, consistent with active faults mapped near the Arauco peninsula (Fig. 1d). The method also captures basal erosion linked to the few subducted seamounts imaged around the Iquique rupture (Geersen et al., 2015) (Fig. 1c).





To interpret the origin of plate interface deformation, we set the limit between accretion and interface deformation for a difference of friction angle of one degree and then searched for the depth distribution of each process (Fig. 3a-b). Results for different thresholds are also presented.

Accretion mostly occurs at depths shallower than 20 km whereas distributed deformation along the plate interface is found at every depth with three favorable peaks: a first one at 15 km depth, a second at 35±10 km depth and a third at 60±5 km depth. Basal erosion due to bathymetric features is known to be efficient at shallow depth (von Huene et al., 2004), but might also occur slightly deeper as testified by subsidence of forearc basins (Clift and Hartley, 2007). Underplating has been observed at shallow depth (Kimura et al., 2010; Tréhu et al., 2019; Bangs et al., 2020), even for seamount moats (Clarke et al., 2018), but mostly occurs below 30-40 km, i.e. the long-term coupling-decoupling transition (Agard et al., 2018). The compilation of P-T conditions for recovered underplated material shows a similar distribution as CTT, with peaks at depths of 30±5 km and 50±5 km and a gap in between (Fig. 3b). Numerical modelling also suggests that erosion predominates over underplating at depths shallower than 20 km (Menant et al., 2020). As a consequence, we assume that the interface deformation documented here mostly relates to basal erosion for depths < 20 km, whereas underplating dominates at greater depths, in particular below the coast.

This depth divide between basal erosion and underplating is strengthened by estimates of pore fluid pressure ratio (Fig. 3d-f). If low pore fluid pressure is mainly found in accretion conditions, higher fluid pressure dominates where basal erosion occurs (von Huene and Ranero, 2003; Vannucchi et al., 2012). On the contrary, recovered underplated material from greater depth present very consistent P-T conditions of lithostatic pressure, which rules out any significant fluid overpressure (Agard et al., 2018). The trimodal distribution deduced from CCT (one peak for basal erosion and two peaks for underplating) and differences in pore pressure remains for a $0.5^o$ or a $2^o$ difference of friction.

Most of the critical segments attesting to distributed interface deformation have lengths smaller than 7 km, and do not exceed 20 km (Fig. 3c). This is in good agreement with the length of graben and horst and seamounts observed on the subducting plate (von Huene and Ranero, 2003; Geersen et al., 2015). This is also consistent with seismic observations (Kimura et al., 2010; Tréhu et al., 2019; Bangs et al., 2020) and the long-term rock record showing generally $\leq$ 300-500 m thick exhumed slices from the top of the slab (Agard et al., 2018) extending 5-10 km downdip at most (Plunder et al., 2012, 2013). Such tabular shape ratios, with seamounts probably near the higher bound (Bonnet et al., 2019), further support interface deformation parallel to the megathrust (Fagereng and Sibson, 2010; Rowe et al., 2013).

## 3.2 Relating plate interface deformation with long-term coastal uplift

In order to approach internal effective friction, the basal friction of the wedge needs to be relatively high (Fig. 2a). Deformation captured at depth by CTT therefore corresponds to underplating in the making, when deformation is still distributed: once tectonic slicing is achieved, basal friction will reach low values characteristic of mature faults, bringing the taper in the stable domain and impeding detection by CTT.

We here show that active, incipient underplating takes place along most of the Chilean margin (Fig. 1, 4a), whether or not strain accumulation ultimately turns these critical zones into tectonic slices (Agard et al., 2018; Bonnet et al., 2019; Kimura



et al., 2007). Along the Tocopilla and Maule segments, incipient underplating takes place below the slab - continental Moho intercept, whereas it is located above and near the intercept between 25-33$^o$S. Along the northern erosive margin (Fig. 4a-b), the thin oceanic sediment cover together with eroded material and possibly pieces stripped from the subducting plate are being

underplated. Southward, larger sediment input favors shallow accretion and underplating of subducted sediments (Fig. 4a-b), as locally observed by seismic surveys (Tréhu et al., 2019).

To evaluate the possible link between coastal uplift and regions of distributed deformation along the plate interface, we compare our spatial distribution of underplating with documented marine terraces and uplift rates (Saillard et al., 2017) (Fig. 4a-c). While the irregular distribution of marine terraces may relate to fragmentary preservation rather than discontinuous

underplating, most of them coincide with segments of underplating (or with standard accretion near the Arauco peninsula). The typical length of the segments showing distributed deformation along the interface is modest (∼10-20 km). This advocates for propagation of deformation only into the topmost part of the subducting plate, e.g. into the stack of oceanic/eroded material or along a hydrated basalt layer within the crust. Although the comparison with uplift rates inferred from morphometric analysis and modelling of landscape evolution (Melnick, 2016) shows no particular pattern, underplating of relatively thin slices would

be consistent with both the low uplift rates and the rock record (Agard et al., 2018).

Shallow normal faults identified along northern Chile have been related to potential underplating (Adam and Reuther, 2000; Clift and Hartley, 2007). We here show that they are located above underplating areas, and that normal faults transition southward to thrusts where accretion prevails (Fig. 4a-c).

### 3.3   Large megathrust earthquakes are surrounded by plate interface deformation

We now compare zones of distributed deformation with areas struck by recent large earthquakes (Fig. 1c). The Mw 8.1 2014 Iquique earthquake is surrounded by patches of distributed deformation associated with basal erosion according to their depth and to seamounts detected in the area (Geersen et al., 2015). The aftershock, located 50 km south of the main event, also falls in between patches of distributed deformation. The Mw 7.8 2007 Tocopilla earthquake is limited up-dip as well as to the south by interface deformation attributed to underplating. The Mw 8.1 1995 Antofagasta earthquake seems limited up- and down-dip by

distributed deformation along the plate interface. The high slip patch of the Mw 8.2 2015 Illapel and the Mw 8.7 2010 Maule earthquakes are both limited down-dip. Their frontal extent partially propagates in areas at critical state but corresponding to standard accretion. A small portion of the areas of limited slip overlaps the distributed deformation patches. However, Fig. 5 shows the diversity of published co-seismic models, emphasizing the uncertainties related to the proposed slip areas, that need to be taken into account for the comparison.

To summarize, along the accretionary part of the margin, mega-earthquakes are delimited down-dip by underplating whereas along the erosive domain, the extent is controlled by both basal erosion, at rather shallow depth, and underplating.

We now investigate the relationships between the critical segments characterized by relatively high effective basal friction, hence expected to behave aseismically, both with the along-strike segmentation inferred for large historical earthquakes (Mw ≥ 7.5, Saillard et al., 2017, Fig. 4a-d) and with the estimates of coupling (Metois et al., 2016; Klein et al., 2018, Fig. 4a-

e). Segments with limited underplating coincide with a significant number of rupture terminations (blue and grey overlays,





respectively: Fig. 4a-d), particularly in the southern region. The distribution of underplating also shows a long wavelength (4 to $8^o$ degrees latitude) corresponding to the three main seismic domains visible on Fig. 4d (respectively located north of Mejillones, between Mejillones and Punta Choros, and to the south of Punta Choros). No correlation exists with sediment input (Fig. 4b). Marine terraces are mostly found where underplating is maximum, i.e. in the central domain (Fig. 4c). The

proportion of a swath profile at critical state is then compared to the average estimate of coupling along this profile (Fig. 4e). A positive correlation is evidenced at small scale (1.5 and 2 $^o$ degrees latitude, Suppl. Mat.) between the segments with minimal underplating (blue overlays; Fig. 4a) and the regions of minimal coupling (red overlays; Fig. 4e). On a larger scale (4 and $8^o$ degrees latitude, Suppl. Mat. fig.7), an anti-correlation is observed with coupling: zones of high coupling correspond to domains where underplating is somewhat more limited (e.g., between 31 and $38^o$S). As shown by Fig. 1d, areas marked by

intense plate deformation are mostly located down-dip of the highly locked patches, in particular in the south. Continuous and voluminous sediment influx provides spatially and temporally stable mechanical conditions (Olsen et al., 2020). This leads to stationary seismic asperities followed, once the Moho is crossed, by the regular underplating of very thin and relatively short slices (typically 300-500 m thick, 5-10 km long). This spatial and temporal stability promotes topographic build-up, explaining the spatial coincidence between strongly coupled patches and efficient underplating. In contrast, segments with

limited underplating might reveal spatially and temporally heterogeneous conditions along the megathrust (Olsen et al., 2020), i.e. heterogeneous deformation impeding earthquake propagation and significant topographic build-up.

Noteworthily, the hypocenters of the last major events lie at the boundary with a patch of distributed deformation. Finally, we show that the high frequency radiations generated by Maule and Illapel earthquakes (Wang and Mori, 2011; Meng et al., 2015, 2018) coincide with areas of extensive interplate deformation (Fig. 1c), along which spatially restricted fault planes

might have been dynamically triggered.

## 4  Discussion - Conclusion

### 4.1  Link between earthquake ruptures and plate interface deformation

We herein show two major results: (1) recent earthquakes are bounded by extensive plate interface deformation characterized, along the southern accretionary part of the margin by underplating at their down-dip edge and, along the northern erosive part

of the margin by both basal erosion and underplating (Fig. 1); (2) along strike, segments characterized by minimal interplate deformation largely coincide with lateral earthquake terminations (circles; Fig. 4d). These observations demonstrate a close link between distributed interplate deformation and earthquake segmentation.

We therefore propose that plastic deformation and stress build-up associated with interplate deformation along distributed fault planes of limited extent (Fig. 6 - step 1), eventually leads to mega-earthquake nucleation (Fig. 6 - step 2). This process may

be accompanied by transient seismic/slip events hardly resolvable by geodetic means due to their size, depth or distance to the coast, as expected in zones of distributed deformation and of high pore fluid pressure (Liu and Rice, 2007; Kimura et al., 2010; Collot et al., 2017). This is suggested by the location of hypocenters at boundaries with distributed deformation (Fig. 1) and concords with observations for the Iquique earthquake (Ruiz et al., 2014a; Meng et al., 2015; Socquet et al., 2017) and others





elsewhere (Tohoku, Kato et al., 2012), (Guerrero, Radiguet et al., 2016). Once nucleated, large earthquakes propagate along
well localized and smoothed rate-weakening fault planes (Bletery et al., 2016) bounded by elongate zones of underplating (or
basal erosion when present; Fig. 1, 6 - step 3). These earthquakes abut on regions of heterogeneously distributed deformation
and stress concentrations (Fig. 6 - step 4), which inhibit the development of well localized slip zones (Wang and Bilek, 2014)
and impede rupture propagation (Wesnousky, 2006). Some of these fractures might however slip seismically during large
events and produce the observed high frequency radiations (Meng et al., 2018, 2015).

## 225   4.2  Down-dip segmentation and plate interface deformation

Within this framework, domain-C (Lay et al., 2012) would correspond to the region of incipient underplating, and domain-
B to the smoothed rate-weakening fault zone. This is consistent with partial coupling of domain-C, where deformation is
dominated by creep mechanisms, and elastic strain is accumulated along undeformed segments of the megathrust, producing
moderate slip earthquakes with a higher recurrence time and significant coherent short-period seismic radiations. Domain-C
might more specifically be delimited by the two underplating peaks (Fig. 3), the upper one corresponding to the slicing of
oceanic sediments and/or eroded continental material, the second to the propagation of deformation into the hydrated basalt
layer once the continental Moho is crossed (Agard et al., 2018), a process ultimately leading to the underplating of very thin
and relatively short slices (typically 300-500 m thick, 5-10 km long, Agard et al., 2018).

Data suggest that domain-C, by locally concentrating interplate permanent deformation due to specific geometry (i.e.,
morphological asperities) and/or mechanical behaviour (i.e., frictional properties, fluid content, porosity), shields the rate-
weakening domain-B from long-lived deformation (Fig. 6). Contrary to the prevalent paradigm, domain-B should not be re-
garded as an 'asperity', i.e. a strongly coupled yet mechanically weak zone with high recurrence time earthquakes, but instead
as a relatively weak zone devoid of permanent deformation and only storing elastic energy. In this new paradigm, loading
along domain-C tightly controls slip deficit as well as rupture nucleation and extent along domain-B (Fig. 6), explaining the
correlation observed between historical earthquakes, earthquake terminations and distributed plate interface deformation (Fig.
4 a,d). On the short-term scale, elastic deformation generates subsidence at the coast during domain-B earthquakes and up-
lift for domain-C events (Melnick, 2016). On the long-term, the thin underplated slices ultimately build a Myr topographic
signal of slow uplift, particularly difficult to capture with geodetic measurements due to the interplay of elastic and plastic
deformation (Menant et al., 2020). The absence of upper plate deformation above the seismogenic zone in conjunction with
topographic build-up above underplating areas account for the correlation between seismic behavior and gravity anomalies
(Song and Simons, 2003).

## 5   Conclusions

We herein show, through a simple CTT analysis of topography, that, along the Chilean subduction zone, earthquakes are
bounded by long-lived, extensive plate interface deformation occurring preferentially at $35 \pm 10$ and $60 \pm 5$ km depth. Along
the Central Chilean margin, earthquakes are limited along their down-dip edge by underplating while, along Northern Chile,
they are surrounded by distributed deformation related to both basal erosion and underplating. Plate interface deformation might control earthquake nucleation, extent and arrest, and its distribution provide a relatively faithful image of the seismic potential at depth. This topography analysis may also remedy the lack of preserved marine terraces or weakly constrained coupling models. This method could therefore be used more systematically to identify regions with precursory transient sig-
nals and to enhance predictions of future earthquakes extent.

**Appendix** See supplementary material for supporting figures.

*Code and data availability.* Code and data available on demand.

*Author contributions.* NC designed the study; NC and RT ran the simulations; NC, RT, PA analyzed the results; NC and PA wrote the paper.

*Competing interests.* Authors declare no competing interest.

*Acknowledgements.* This project has been funded by French National Research Agency (ANR) grant SEAFRONTTERA ANR-13-PDOC-0013–01.





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



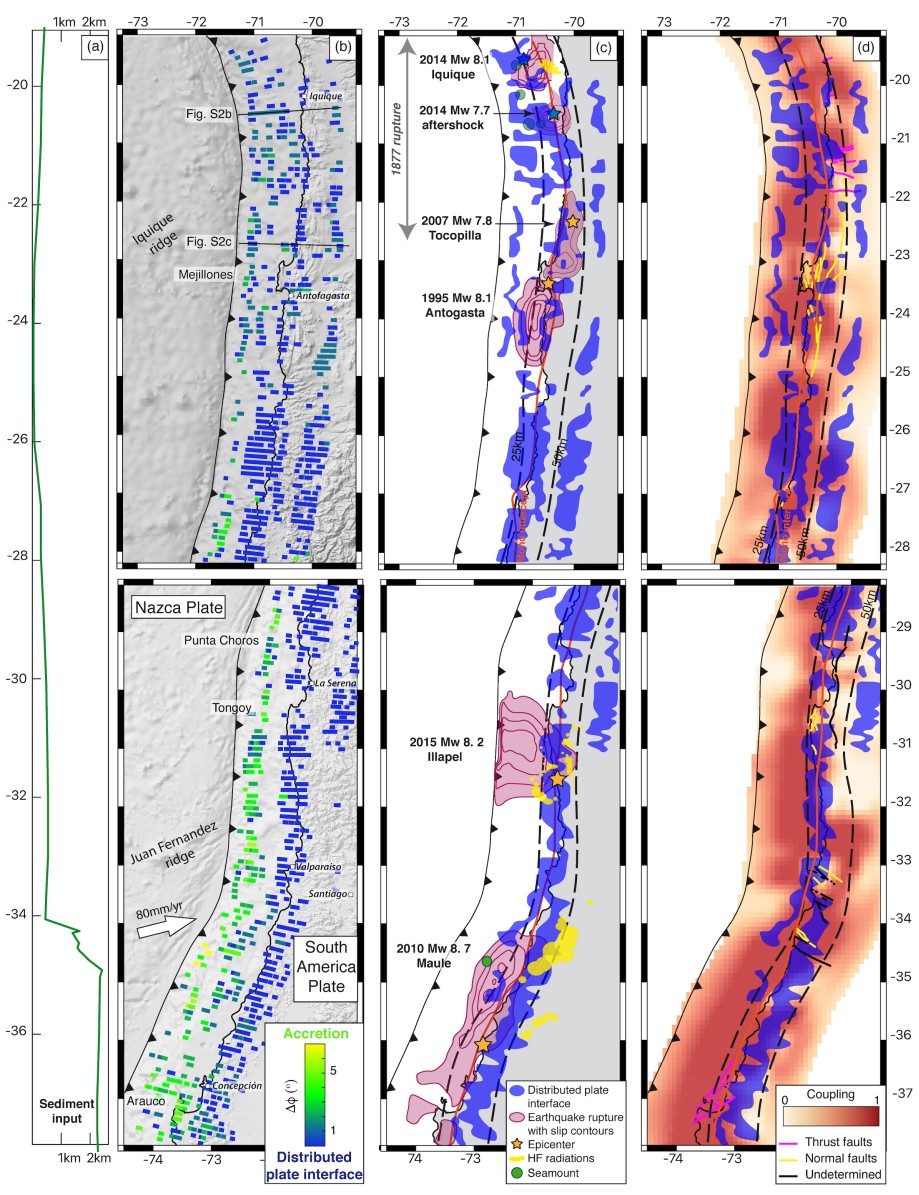

**Figure 1.** (a) Sediment thickness at trench (Santibáñez et al., 2018; Tréhu et al., 2019). (b) Difference between internal and basal effective frictions of identified critical segments. Large differences lead to standard accretion with thrust faults reaching the surface, while small differences lead to the development of faults parallel to the plate interface allowing for either basal erosion or underplating. (c) Interpreted patches with distributed deformation along the plate interface related to either basal erosion or underplating compared to last major events (Maule slip contours every 2.5 m (Lin et al., 2013); Iquique 2014 (Ruiz et al., 2014b), Illapel 2015 (Tilmann et al., 2016), and Antofagasta 1995 (Chlieh et al., 2004) from 1 m every meter; Tocopilla 2007 (Béjar-Pizarro et al., 2013) and aftershock of Iquique 2014 0.5 m slip contours (Ruiz et al., 2014b)). Orange stars: gCMT solutions, blue stars: CSN catalog. High frequencies radiations (yellow, Wang and Mori, 2011; Meng et al., 2015, 2018), identified seamounts (green, Geersen et al., 2015; Maksymowicz et al., 2015), slab depths (dashed black, Hayes et al., 2018) and slab-Moho intercept (orange, Tassara and Echaurren, 2012). (d) Interpreted patches with distributed plate interface compared to coupling model (Metois et al., 2016) with known active faults (Santibáñez et al., 2018).





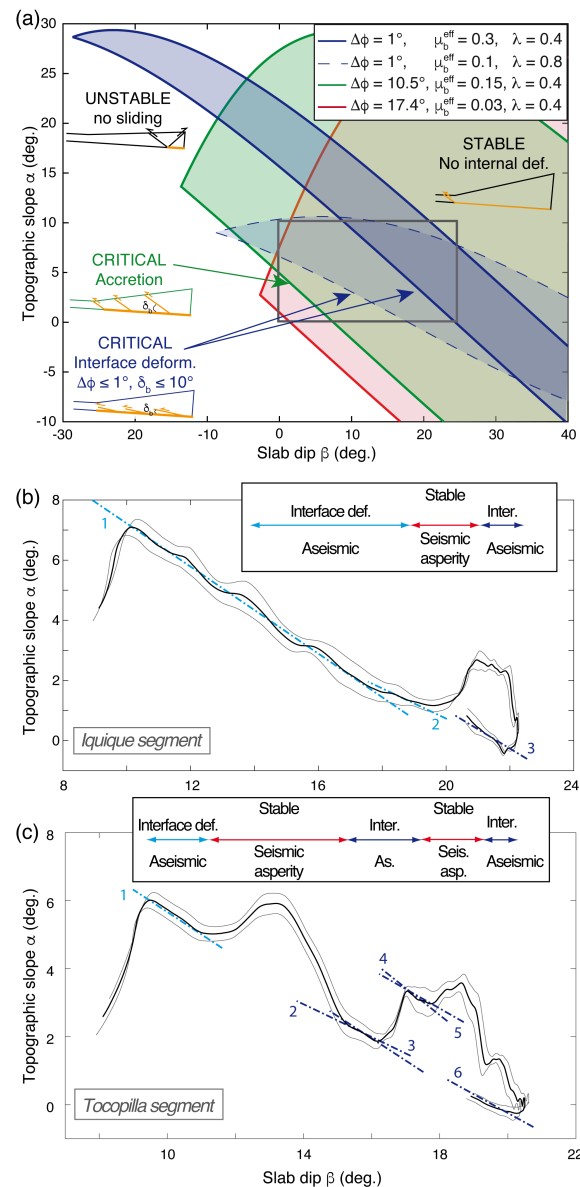

**Figure 2.** (a) Critical taper theory with different mechanical states. The grey rectangle represents the standard range of topographic slope $\alpha$ and slab dip $\beta$ of subduction zones. With extremely low effective friction along the megathrust (red curve), the wedge systematically falls in the stable domain. With larger effective friction, and large difference of effective friction between the megathrust and the wedge (green curve), the wedge can reach the accretionary critical state characterized by internal faulting. With a reduced difference (blue curves), a wedge can still reach the critical state but the dip of internal faults within the wedge will decrease and become parallel to the plate interface. High pore fluid pressure facilitates interface deformation for low slab dips (dashed blue curve), whereas lower pore fluid pressure induces interface deformation at larger slab dip (plain blue curve). (b) Topographic slope ($\alpha$) versus slab dip ($\beta$) for a swath profile along the Iquique segment and (c) along the Tocopilla segment (locations on Fig. 1b and Suppl. Mat. fig. 1). Segments at critical state, according to inversion, are shown in blue: light blue when probably erosive, dark blue for probable underplating. Grey: swath plus or minus standard deviation. Properties of each segment are provided in Suppl. Mat. table 1.





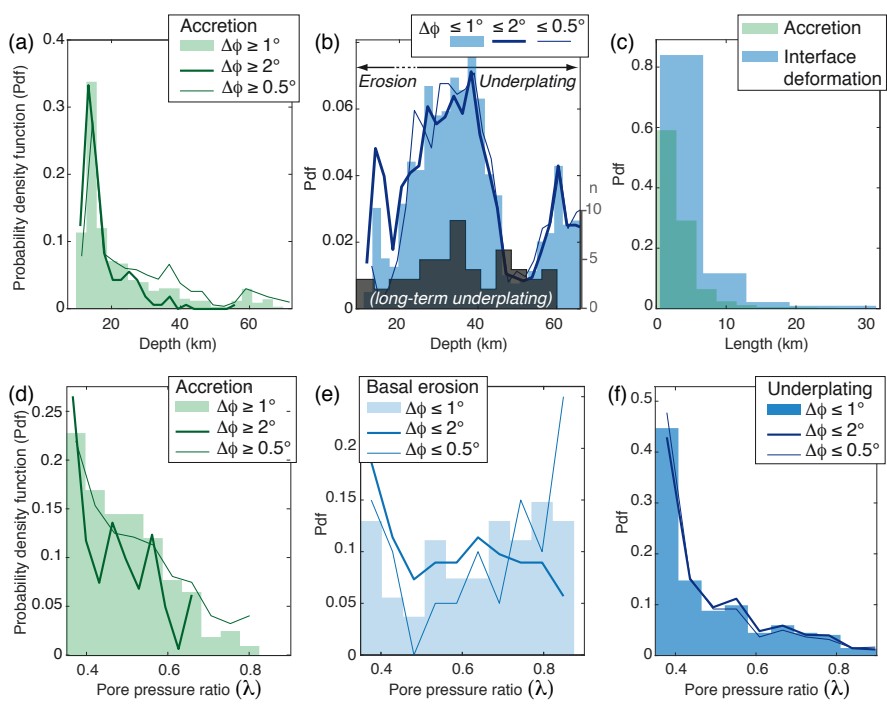

**Figure 3.** (a) Mean depth of segments at accretionary critical state and (b) distributed plate interface deformation critical state, compared to a compilation of the maximum burial depth reached by exhumed tectonic slicing (grey, Agard et al., 2018) (c) Length of segments at accretionary and distributed plate interface deformation critical state. (d), (e) and (f) Pore fluid ratio $\lambda$ of segments at accretionary, basal erosion or underplating critical state. Histograms are shown for a $1^o$ difference of internal and basal effective friction, thin line for a $0.5^o$ difference, bold line for $2^o$.



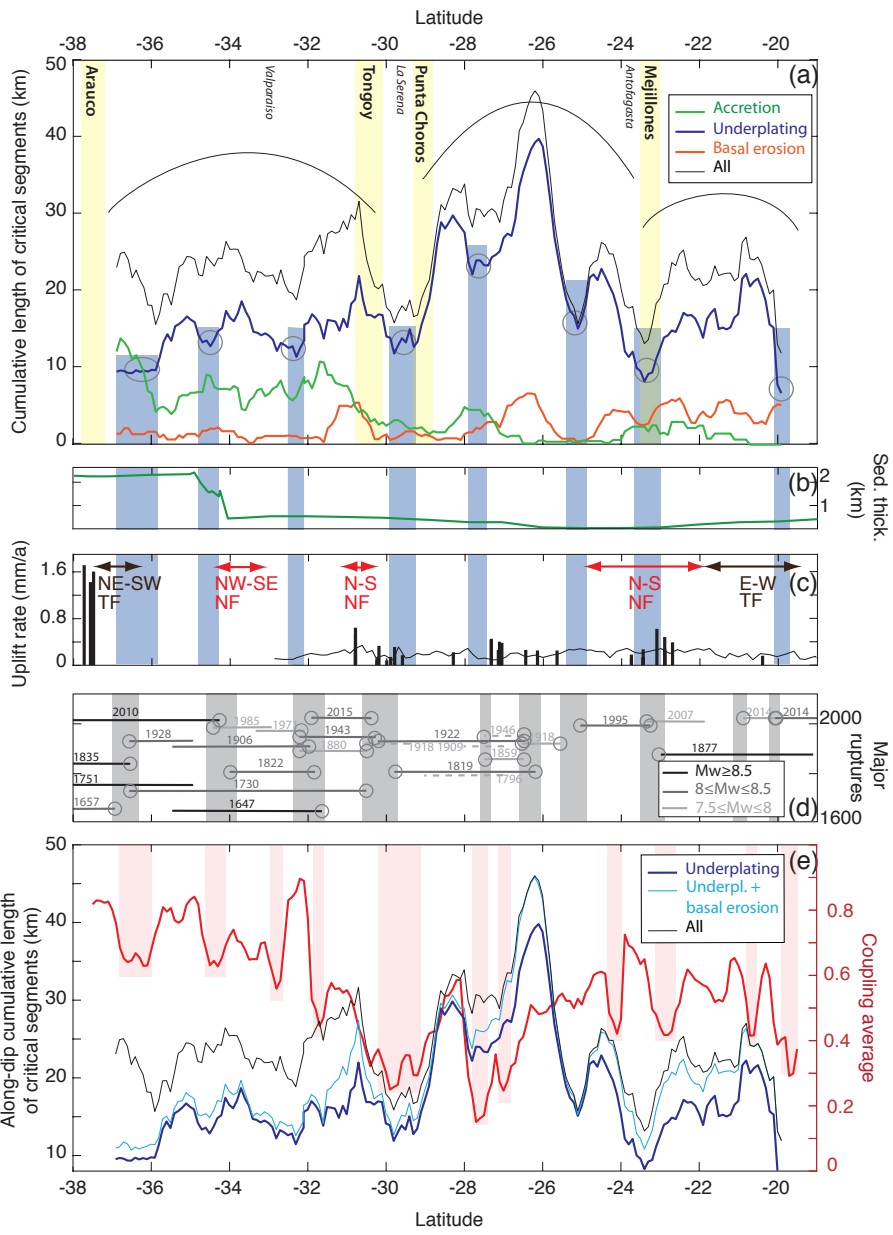

**Figure 4.** (a) Along-dip cumulative length of segments at accretionary (green), underplating (blue) or basal erosion (orange) critical state (length smoothed by a rectangular window over 8 profiles). Regions with limited underplating length are represented by blue bands. (b) Thickness of sediment income (green, Santibáñez et al., 2018; Tréhu et al., 2019). (c) Quaternary uplift rates of marine terraces (black bars, largest average rate since terrace abandonment, Saillard et al., 2017), coastal uplift rate obtained from a landscape evolution model (black curve, Melnick, 2016) and extent of known active faults (NF: normal fault, TF: thrust fault, Santibáñez et al., 2018). (d) Along-strike extent of historical megathrust ruptures (Saillard et al., 2017). (e) Comparison between the cumulative length of segments at critical state (thin black), with underplating (thick blue), with both underplating and basal erosion (light blue) and the average coupling value along profiles from Metois et al. (2016) corrected by Klein et al. (2018) between $24.5^o$S and $28^o$S (plain red curve). Red bands show segments of decreasing coupling. Correlation factor between coupling and underplating: $r = -0.36$, between coupling and underplating + basal erosion: $r = -0.39$.



**Figure 5.** Areas prone to distributed deformation along the plate interface compared to various published co-seismic slip models (Duputel et al., 2015; Schurr et al., 2012; Pritchard and Simons, 2006; Tilmann et al., 2016; Vigny et al., 2011; Ruiz et al., 2014b; Chlieh et al., 2004; Moreno et al., 2010; Hayes et al., 2014; Béjar-Pizarro et al., 2013; Lin et al., 2013). Orange stars: gCMT solution, blue stars: CSN solution.





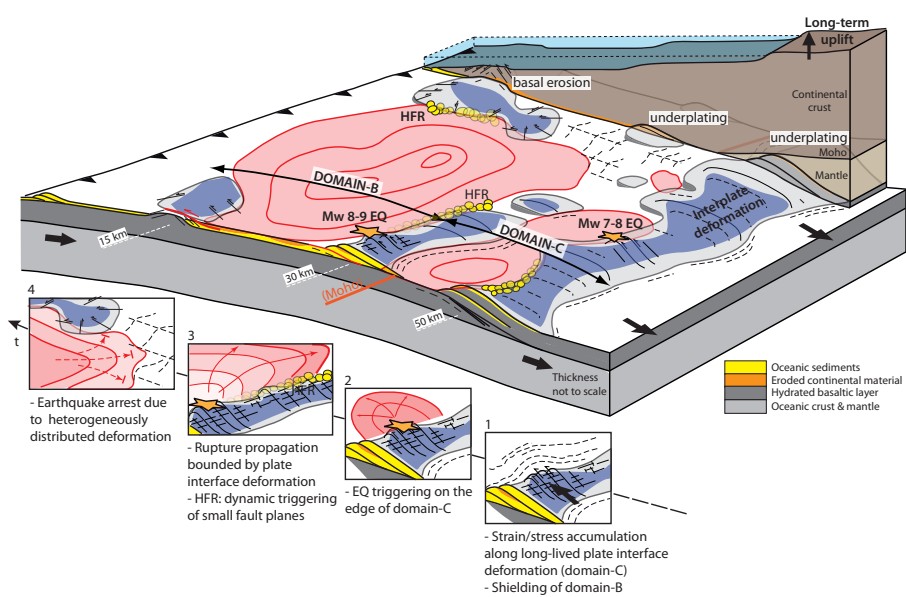

**Figure 6.** 3D view onto the subduction plate interface depicting the mutual spatial relationships between earthquake ruptures and longer-lived deformation. Stages 1-4 illustrate how this deformation controls the triggering and limits seismic ruptures. See text for further discussion.