# Peer review of "Earthquake ruptures and topography controlled by plate interface deformation"

_Solid Earth, 2021_

## Author Comment (AC1)

Dear Reviewer,

We thank you for your helpful comments.

We have favorably answered to all the comments (see below).

Nadaya Cubas on behalf of the co-authors.

**Reviewer 1:**

Cubas and coauthors present a study that focuses on interplate deformation along the Chilean subduction zone. Applying the critical taper theory, Cubas and coauthors are capable of mapping plate interface deformation. Then, they compare it with a series of long-term and short-term observables of subduction dynamics including coastal uplift, historical record of large interplate earthquakes and geodetically derived coupling. At the end of the manuscript the Authors provide a conceptual model where large megathrust earthquakes are triggered and stop in correspondence of areas of interplate deformation. This model is crucial for seismic hazard assessment as it can be potentially used for mapping areas of the megathrust that might be prone to hosting large earthquakes in the future. The subject is of course very interesting.

This paper will be of interest for the structural geologists, and geoscientists specialized in tectonics, geodynamics and seismology and I think it will be a valuable contribution for SE.

The analysis is based on the critical taper theory. The Authors use an inversion procedure that allow them to retrieve friction of the wedge, the pore pressure ratio of the wedge and the effective friction of the megathrust as described in an earlier paper.

The manuscript is well structured and clearly written. Figures are clear and supporting material is helpful for complementing this study. Motivation, procedures and quantitative analysis are clearly explained.

I have only one minor concern and few technical corrections reported below.

**Specific comments**

Authors claim that "Once nucleated, large earthquakes propagate along well localized planes". A reader can see that for the vast majority of the cases this is true but there are also some exceptions (independently from the slip model). Slip maps of Antofagasta, Maule and Iquique show some degree of overlap with areas of distributed deformation. Even near the location of highest slip as for Maule… This observation is against the model shown in figure 6. I suggest to add a few sentences in the discussion and/or in paragraph 3.3 where Authors justify the existence of such exceptions.

In paragraph 3.3, we have added:

***A small portion of the earthquake ruptures do overlap with the plate interface deformation, but high slip areas are always clearly limited by plate interface deformation***.

In the discussion:

*Once nucleated, large earthquakes propagate along well localized and smoothed rate-weakening fault planes \citep{bletery2016mega} limited by elongate zones of underplating* **which, in addition, inhibit further rupture propagation and slip** *(Fig. \ref{fig:1}, \ref{fig:5} - step 3).*

**Technical corrections**

Title: Consider adding information about the study area and/or the subduction environment.

Done.

In the abstract specify that the word "prediction" (line 2 and 16) indicates the location and extent (as correctly done in the introduction).

Done.

Line 100 "the inversion procedure" sentence is a repetition of the first few words of that same paragraph (line 93).

This paragraph has been reworked.

Line 175 "The high slip patch of the Mw 8.2 2015 Illapel and the Mw 8.7 2010 Maule earthquakes are both limited down-dip". Limited by what?

*Plate interface deformation*, added.

Line 186: "seismic domain" consider using another word to prevent confusion with "domain" proposed by T. Lay for depth. Segment might work?

Ok, changed for *region*.

Line 187: figure 4a instead of 4d, right?

Corrected.

Doublecheck title of paragraph 4; i.e., delete "conclusion"

Corrected.

Figure 2: I find very helpful the colorcoding as in Figure s4. Consider modifying panels b and c.

We have added cross-sections to help readers instead.

Figure 4. It is important to avoid reader skepticism: how do blue bands in panels a-c are defined? They appear as local minima but criticism might rise from the fact that the blue band around 28 latitude is way higher than peaks (e.g., Tongoy). Be sure Authors specify underlying assumptions.

True, we now specify:

*Segments with limited plate interface deformation ($<$15\,km) coincide with a significant number of rupture terminations (blue and grey overlays, respectively: Fig. \ref{fig:4}a-d),*

*particularly in the southern region.  Between 29 and 24$^o$S, rupture terminations coincide with segments where plate interface deformation is reduced.*

---

## Author Comment (AC2)

Dear Reviewer,

We thank you for your detailed and helpful comments.

Your main comment concerned the too condensed and brief explanation of the methodology. Since the inversion procedure was already published (Cubas et al., EPSL 2013), we thought that these details were unnecessary. But, we decided to follow your advices. In the main text, we now provide more carefully the inputs, the explored parameter space, the criteria for the segment selection and the misfit cutoff value. The whole procedure and equations are now provided in the Supplementary Material. Also, to improve the readability of the methodology section, figure 2 was splitted in 2. The figure has been reworked as advised to facilitate the understanding of the inversion procedure.

The second main comment concerned our chosen threshold to separate basal erosion and underplating (20km depth). We would like to remind that this threshold was based on the literature, as stated in the main text from lines 138-146. The assumption was clearly stated in the text (*we **assume** that the interface deformation documented here **mostly** relates to basal erosion for depths < 20 km, whereas underplating **dominates** at greater depths, in particular below the coast.*). However, following your advice, we now more carefully insist on the fact that this threshold is an assumption throughout the text (lines 151-152; 198; 202-205; captions fig. 3, 4 & 5). Moreover, on figure 4 (now 5), we now also compare the rupture extent with the distributed deformation, to show that the conclusion stands even without making any assumption on the origin of the plate interface deformation (basal erosion or underplating).

We also made all the corrections proposed for the figures and we favorably answered to all the minor comments when adequate (see below).

We hope that all of these improvements will meet your expectations.

Nadaya Cubas on behalf of the co-authors.
* * *
**Reviewer 2:**

In the present manuscript, Cubas et al. estimate the distribution of different forms of plate interface deformation (accretion, basal erosion, underplating) along the Chilean margin using critical taper theory, and compare their findings to the rupture extents of large earthquakes, regional uplift and interplate coupling. The subject is highly interesting, and the findings provide some reason to believe that plate interface deformation indeed plays a very important role in shaping the long-term and short-term behavior of the plate boundary.

The manuscript is well-written and illustrated and fits very well into the scope of the journal. My main concern about the manuscript is that it was apparently written for a shorter-format journal and not adapted (much) before submission here, so that it is unnecessarily brief and condensed in places, which negatively affects readability and clarity. Moreover, in some places choices that were made need to be better explained (or explained at all) since they currently appear like "magic", and it is not always clear if the presented correlations critically depend on said "magic" or not. I thus recommend moderate revisions and will describe these concerns in more detail below, before presenting less important specific comments by line number.

**General comments:**

I think the authors should invest some time to transform their manuscript into a longer-format version that is less condensed and thus easier to read and comprehend. As is, I find some of the figures to be too complex (sometimes unnecessarily so; see comments to Figures below), and some important issues are not or only very tersely described. It would also be nice if the manuscript tried to go step by step and separated observations from interpretations (first describe results, then interpret). Examples of where more detail would be nice to have will be provided below.

In some places, there needs to be a better (or any) explanation of how things were actually done. For instance, Figure 4 correlates the lengths of determined segments in accretion, basal erosion and underplating with other parameters (interplate coupling, rupture extents of historical earthquakes etc.). While the principle of how the segments were obtained is shown in Figure 2c and d, what lacks is a clear description of when such segments were kept or discarded. The text only mentions that only segments with "extremely low misfits" were kept, without mentioning what that means or even how the misfit was defined in the first place. Moreover, a relatively arbitrary depth threshold of 20 km is the only parameter that separates basal erosion from underplating. I think such critical choices and procedure need more in-depth explanation, otherwise it is very hard to judge whether correlations like the ones shown in Figure 4 are robust.

Also, the utilized inversion procedure is kept completely in the dark, and it is not clear how the probability density distributions shown in Figure 3 are obtained. I understand that the methodology has already been published elsewhere, but at least a short summary of what are inputs, tuning parameters and outputs is necessary in my view.

As stated above, we now more carefully describe the inversion procedure, and how these segments were obtained is now made clearer (method and misfit threshold).

We also show that results of figure 4 do not change when no assumption is made on the origin of the interplate deformation.

We also now systematically recall throughout the text that the 20km depth is an assumption (based on literature).

And Figure 3c has been corrected thanks to reviewer advice.

**Specific comments:**

Title: it should mention that this is a study of the Chilean margin

We agree, and changed.

l.2: "earthquake ruptures" should maybe be replaced by "earthquake rupture extents", to make clear that prediction here only refers to the size and location, not to the time of occurrence

Done

l.3/4: seismic and aseismic patches is a bit unclear, maybe better say strongly and weakly coupled? If one looks at the interseismic period, strongly coupled patches ("seismic" because they produce large earthquakes) show nearly no microseismicity, whereas weakly coupled

regions ("aseismic" because they don't produce large earthquakes) show a constant background of small events

We agree and changed

l.8ff: better to say south and north of 35 degrees S (or S and N of where the Juan Fernandez Ridge is subducted). Even the southern termination of the study area is usually referred to as Central (sometimes South-Central) Chile

True, done.

l.10: "all major earthquakes" here refers to two events (Illapel and Maule), correct?

Right, corrected for the two last major.

ll.26ff: one could add pore fluid pressure variations (e.g. Moreno et al., 2014, NGeo) and plate interface geometry (e.g. Bletery et al., 2016, Science).

From our point of view, pore fluid pressure belongs to frictional properties, and plate interface geometry to plate roughness. But we changed for:

Subduction earthquake propagation has so far been either related to megathrust frictional *and pore fluid pressure* properties (perfettini, et al; 2010; Kaneko et al., 2010; *Moreno et al., 2014*) … or seafloor roughness (kodaira et al., 2000; Wang et al., 2014; *Bletery et al., 2016*).

l.32: remove "the" before million years

done

l.34: also elsewhere, see e.g. Malatesta et al. (2021, JGR)

Right, as in Ruff & Tichelaar 1996, but the paragraph here is dedicated to Chile.

l.46: this may be true for single seamounts, but there are quite a few correlations of rupture extents with larger incoming seafloor features such as ridges or fracture zones. Both ends of the Illapel earthquake rupture can, for instance, be associated with such features (Challenger Fracture Zone and Juan Fernandez Ridge).

Right, as we wrote lines 42-47. In this sentence, we present papers that have imaged this correlation from seismic surveys after the earthquake occurrence.

l.60: Unclear what "To do so" refers to...better leave out.

Done.

l.94: is latitudinal degree meant? or every 0.1 degrees along-trench? Longitudinal does not make sense, since the margin is nearly north-south

We changed for: *perpendicular to the trench every 0.1 degree along trench*.

ll.93-105: this paragraph needs extension and clarification; it is not clear to me what exactly is done. The authors should give a quick summary of what the utilized inversion approach does, what are the inputs (only bathymetry/topography and slab dip or is there more?) and assumptions/parameter choices. What is this rectangular window that is used for smoothing (window length, how is smoothing done; also that a triangular window has been tried out in Figure S5 is not even mentioned here), and how are its parameters chosen? I also don't understand what segments parallel to critical envelops are (this becomes somewhat clear when looking at the Supplementary figures, but needs to be at least briefly described here...also, how was this selection done, visually or automatically; if the latter what were the criteria?).

In the main text, we now provide more clearly the inputs (alpha, beta), the parameter space, the criteria for the segment selection, the misfit value cutoff and the parameters cut-off. The whole inversion procedure is described in the Suppl. Mat.
To go step by step, as advised by the reviewer, Figure 2 was splitted in two.

ll.101/102: this means that results falling outside this range were simply discarded? Also, what are extremely low misfits? I recommend being clearer here.

Yes, as now explained. Extremely low misfit is lower than 0.1, value now provided.

l.113: I think 35 degrees S is meant

Rather north of 26 degrees S, when sediment income is equal to 0 (see figure 1).

l.119: what is the motivation/reasoning for this choice? Also, should this not be mentioned previously (i.e. before the previous paragraph), where erosion and accretion have already been interpreted?

The motivation is argued from lines 138-146. Where erosion and accretion occurs is mentioned in the introduction, lines 56-57, and represented on figure 1a with the sediment thickness.

l.122: It would make sense to put these depths into relation with the depth of the continental Moho right here (maybe also show the range of continental Moho depths in the histograms of Figure 3a,b)

A boxplot of the Moho intercept depth has been added to figure now 4a and b.

l.137: could the first peak for basal erosion in Figure 3e, the one at low pore fluid pressures, be there because the assumption of 20 km separating basal erosion and underplating is not perfect, and some underplating at shallower depth is mapped into the basal erosion plot?

It is possible, we have added this remark: *The first peak for basal erosion at low pore fluid pressures in Fig.4e could be related to the 20km depth assumption, and reveal the minor occurrence of some underplating at shallower depth.*

ll.138ff: The length of segments critically depends on the fitting that was done (see comment above), which is not described in detail. I thus find it hard to judge whether such a property can be robustly compared with structures in nature

We do not agree, it was already discuss in Cubas et al., 2013, and this is what figure S6 supports. The inversion procedure is now provided with details.

l.153: mention that this intercept is shown in Figure 1c and d as an orange line

Done.

l.159: I fail to see anything systematic in Figure 4c

We removed the sentence.

l.175ff: According to some recent studies (Schurr et al., 2020, GRL; Sippl et al., 2021, JGR), interseismic microearthquakes along the plate interface may surround the later ruptures of large earthquakes, which means they occur in the same regions that emerge as featuring distributed deformation here. Could this seismicity be a fingerprint of distributed deformation, or would the involved processes be independent from each other?

Of course! This is one of the messages we had tried to convey, but possibly insufficiently clearly. We have now clarified this point and added these references in the discussion:
*We therefore propose that plastic deformation and stress build-up associated with interplate deformation along distributed fault planes of limited extent (Fig. \ref{fig:5} - step 1) eventually leads to mega-earthquake nucleation (Fig. \ref{fig:5} - step 2).*
***This interseismic deformation might be accompanied by some micro-seismicity, as observed prior to the Iquique rupture \cite{schurr2020forming} and in the upper plate in northern Chile \cite{sippl2018seismicity}***

l.186: word missing? (a long wavelength what?)

We changed for: The *underplating segments also show a long wavelength distribution.*

l.210: this sounds as if the results actually discriminate between basal erosion and underplating, but in truth this is an assumption (one occurs at depths shallower than 20km, the other deeper)

In this sentence, we say: *recent earthquakes are bounded by extensive plate interface deformation characterized along the northern erosive part of the margin by both basal erosion and underplating.*
We do not pretend to discriminate between basal erosion and underplating.
We can't for sure locate where basal erosion or underplating takes place. But we know from independent and numerous published studies that basal erosion occurs (e.g, Von Huene and Ranero, 2003; Clift and Vannucchi, 2004; Clift and Hartley, 2007; Geersen et al., 2015), and that the deep interface deformation found here can be attributed to underplating (e.g., Clift and Hartley, 2007).

l.211: while this correlation between earthquake terminations and regions of decreased plate interface deformation is indeed apparent from Figure 4a,d, it would be wortwhile to mention that there are some exceptions to this (e.g. the 1617, 1730 and 1906 earthquakes in Central Chile appear to have ruptured clearly across such regions)

We agree, and that is why we said 'Segments with limited underplating coincide with a ***significant*** number of ruptures terminations'. (We could also discuss the confidence in estimating the lateral extent of earthquakes occurred during the 17th century.)

l.218: these studies indicate that the transient slip events ARE resolvable with geodetic means...it is just a matter of further developing detection approaches (the data are clearly good enough)

ok, we removed this part of the sentence.

l.243: Could the Coastal Cordillera, present along much of the Chilean margin, be a consequence of such a slow uplift due to underplating?

Of course, added in the text: *The Coastal Cordillera present along much of the Chilean margin would result from this long-term underplating process*.

l.252: provides; "faithful" is not a good word here (robust?)

Corrected to 'reliable', in the sense of 'valid', to be 'trusted'.
Note that an image cannot really be 'robust', unless one means high-resolution, perhaps.

*Code and data availability*: I recommend that the authors make the obtained data (i.e. the segments with the angle difference shown in Figure 1) available through a repository. Keeping all results closed is how science worked in the past, nowadays datasets should be open so that others can use and also validate them.

We do agree, the table and code were available for reviewers during the review process, through a google drive link. They will be released after publication.

*Figures:*

Figure 1: The profiles shown in the upper panel of subfigure b are mislabeled, since they are not shown in Figure S2 but in Figure 2b,c. Those profiles that are shown in Figure S2 (in the southern part) should also be marked with their locations here.

Corrected.

I do not like that different extents and contours are used for the different earthquake ruptures (2.5 m for Maule, 1 m for most others, 0.5 m for Iquique aftershock and Tocopilla)...a consistent value should at least be used for the extent of the pink regions, otherwise this leads to big distortions in terms of the represented rupture area

Contour slip values have been added, and contours have been harmonized for equivalent magnitude earthquakes.

Figure 2: it would be nice if there could be a map view subfigure next to subfigures b and c where the position of the different stable and interface deformation segments could be shown. As the graphs are slab dip against topographic slope, there is no way to know which part of the plate interface is represented where (Figure S4 helps here - should be mentioned in the caption).

Agreed. We added a crossection of the swath profile to locate the critical segments identified on figure 2 (figure 3 now).

Figure 3: I find these plots difficult to read. Why are line representations and bar representations of histograms mixed?

We tried superimposing three curves, superimposing three histograms, but these two representations are illegible, and we think it is important to keep the superimposition for comparison. This is why we prefer the keep this representation. We have now added a sentence to the caption to explain our choice (*curves were chosen for sake of clarity*).

I also do not see the use of subfigure c, or why it is placed in a position that implies a close relation to a and b.

Space issue, there is no relation with a and b, we added rectangles in the figure to separate the different parameters.

For subfigures e and f, please mention in the caption that a fixed depth limit of 20 km is assumed to separate basal erosion (at shallower depth) from underplating (at deeper depth). At the moment this is hidden in the text and not mentioned in the caption.

Agreed, it was added to the caption (*assuming that basal erosion prevails above 20 km depth, and underplating below*).

Moreover, I'm rather confused by subfigure c. I assume the probability density functions (Pdfs) are normalized so that the integral over them is 1 (also this should maybe be mentioned in the caption). If so, then how can one curve (interface deformation) be above the other (accretion) everywhere, unless the shown Pdf is truncated and there are significant values that are not shown? Or were massively different bin sizes used (if so, why? And it needs to be mentioned)?

The probability density functions are normalized, the integral is 1. The number of bins are the same (10), the bin lengths were different, it is now corrected.

Figure 5: Are the earthquake slip patches again using different values for Maule compared to the other earthquakes, as in Figure 1? This should at least be mentioned in the caption then.

Contour slip values have been added, and contours have been harmonized for equivalent magnitude earthquakes.

Figure 6: Can the caption briefly explain what pink and blue areas on the megathrust are?

Agreed and added.